# The Occurrence of Hyperactivated Platelets and Fibrinaloid Microclots in Myalgic Encephalomyelitis/Chronic Fatigue Syndrome (ME/CFS)

**DOI:** 10.3390/ph15080931

**Published:** 2022-07-27

**Authors:** Jean M. Nunes, Arneaux Kruger, Amy Proal, Douglas B. Kell, Etheresia Pretorius

**Affiliations:** 1Department of Physiological Sciences, Faculty of Science, Stellenbosch University, Stellenbosch 7602, South Africa; 18131018@sun.ac.za (J.M.N.); arneaux@drakruger.co.za (A.K.); 2PolyBio Research Foundation, 7900 SE 28th ST, Suite 412, Mercer Island, WA 98040, USA; amy.proal@gmail.com; 3Department of Biochemistry and Systems Biology, Institute of Systems, Molecular and Integrative Biology, Faculty of Health and Life Sciences, University of Liverpool, Crown St., Liverpool L69 7ZB, UK; 4The Novo Nordisk Foundation Centre for Biosustainability, Technical University of Denmark, Building 220, Chemitorvet 200, 2800 Kongens Lyngby, Denmark

**Keywords:** myalgic encephalomyelitis/chronic fatigue syndrome (ME/CFS), platelets, fibrinaloid microclots, hypercoagulability

## Abstract

We have previously demonstrated that platelet-poor plasma (PPP) obtained from patients with Long COVID/Post-Acute Sequelae of COVID-19 (PASC) is characterized by a hypercoagulable state and contains hyperactivated platelets and considerable numbers of already-formed amyloid fibrin(ogen) or fibrinaloid microclots. Due to the substantial overlap in symptoms and etiology between Long COVID/PASC and ME/CFS, we investigated whether coagulopathies reflected in Long COVID/PASC—hypercoagulability, platelet hyperactivation, and fibrinaloid microclot formation—were present in individuals with ME/CFS and gender- and age-matched healthy controls. ME/CFS samples showed significant hypercoagulability as judged by thromboelastography of both whole blood and platelet-poor plasma. The area of plasma images containing fibrinaloid microclots was commonly more than 10-fold greater in untreated PPP from individuals with ME/CFS than in that of healthy controls. A similar difference was found when the plasma samples were treated with thrombin. Using fluorescently labelled PAC-1, which recognizes glycoprotein IIb/IIIa, and CD62P, which binds P-selectin, we observed hyperactivation of platelets in ME/CFS hematocrit samples. Using a quantitative scoring system, the ME/CFS platelets were found to have a mean spreading score of 2.72 ± 1.24 vs. 1.00 (activation with pseudopodia formation) for healthy controls. We conclude that ME/CFS is accompanied by substantial and measurable changes in coagulability, platelet hyperactivation, and fibrinaloid microclot formation. However, the fibrinaloid microclot load was not as great as was previously noted in Long COVID/PASC. Fibrinaloid microclots, in particular, may contribute to many ME/CFS symptoms, such as fatigue, seen in patients with ME/CFS, via the (temporary) blockage of microcapillaries and hence ischemia. Furthermore, fibrinaloid microclots might damage the endothelium. The discovery of these biomarkers represents an important development in ME/CFS research. It also points to possible uses for treatment strategies using known drugs and/or nutraceuticals that target systemic vascular pathology and endothelial inflammation.

## 1. Introduction

Myalgic encephalomyelitis/chronic fatigue syndrome (ME/CFS) is a debilitating, multisystem disease that currently lacks a definitive diagnostic biomarker, an effective treatment, and a clear and widely accepted aetiological explanation. Symptoms of the condition include pathological fatigue (uninfluenced by exertion) that goes unresolved with rest and sleep; other symptoms include sleep perturbations, joint and muscle pain, intolerance towards exercise, headaches, gastrointestinal issues, flu-like symptoms, and cognitive impairments [1,2]. The prevalence figures of ME/CFS are blurred by inconsistent case definitions, self-reporting assessments, and a large proportion of undiagnosed patients [2,3,4]. Regardless of these issues, it has been estimated that there are 1.5 million individuals suffering from ME/CFS (low-end value) in the USA alone [5]. Alarmingly, the latest predictions suggest that the prevalence of ME/CFS might face a period of rapid growth, possibly by as much as a factor of six [6].

Most cases of ME/CFS begin with a viral infection or involve multiple exposures to viral and/or bacterial pathogens over time [7,8,9,10]. Viruses implicated in initiating or exacerbating the ME/CFS disease process include human herpes virus (HHV)-6, HHV-7, Epstein-Barr virus (EBV), cytomegalovirus (CMV), enteroviruses, human parvovirus B19, and coxsackie B virus [11]. With regards to bacteria, gut dysbiosis and gram-negative LPS have been suggested to play a role in ME/CFS pathology [12,13,14]. Patients have also met the diagnostic criteria for ME/CFS after infection with bacterial pathogens, including Coxiella burnetii (Q fever) or those in the genus Brucella (brucellosis) [15,16,17].

More recently, 10–30% of patients infected with the SARS-CoV-2 virus during the COVID-19 pandemic have developed chronic symptoms that overlap greatly with those of ME/CFS. These patients are being given the diagnosis Long COVID or Post-Acute Sequelae of COVID-19 (PASC) [18]. The overlap between Long COVID/PASC and ME/CFS symptoms is so profound that many Long COVID/PASC patients meet the diagnostic criteria for ME/CFS after 6 months of ongoing symptoms [19,20,21,22,23,24]. 

We have demonstrated that Long COVID/PASC platelet-poor plasma (PPP) contains large anomalous fibrin/amyloid deposits, which we have named fibrinaloid microclots, and hyperactivated platelets [25]. The fibrinaloid microclots are resistant to fibrinolysis even after trypsinization. When solubilized in the laboratory via a second trypsinization step, it was shown that they contain inflammatory molecules and clotting factors including α(2)-antiplasmin (α2AP), various fibrinogen chains, and serum amyloid A (SAA). This pathology may lead to significant endothelial inflammation, (temporary) capillary blockage, and hypoxia [26].

Due to the substantial overlap in Long COVID/PASC and ME/CFS symptoms and etiology, it is very likely that coagulation-based pathology may also contribute to non-SARS-CoV-2 onset ME/CFS (cases initiated or exacerbated by other pathogens). Despite some inconsistencies in the literature [27,28], there is precedent for hypercoagulability and platelet activation in ME/CFS. Brewer et al. found that ME/CFS individuals presenting with active HHV-6 infection exhibited a state of hypercoagulability [29], although over 80% of subjects involved in the study possessed hereditary risk factors for thrombosis, thereby skewing the precision of this interpretation. Another study demonstrated a hypercoagulable state with platelet activation and suggested that fibrin deposits in microcirculatory vessels (perhaps fibrinaloid microclots?) can adhere to the endothelial lining and play a part in the manifestation of ME/CFS symptoms, possibly by impairing oxygen and nutrient delivery to tissues [30]. More recently, platelet activity and markers have been implicated in ME/CFS, although the authors indicated a loss of significance after statistical corrections [31]. Endothelial abnormalities have also been noted [32,33]. Furthermore, healthy endothelial cells exposed to plasma from ME/CFS individuals exhibited functional defects [34]. 

There are clear molecular mechanisms by which viral/bacterial infection and/or chronic inflammation may contribute to coagulation-based sequelae in ME/CFS. Platelets can sense and bind to both viruses and bacteria via a variety of platelet receptors, and dozens of viral and bacterial products stimulate platelets, modulating their function [35,36,37,38]. If these infections or associated inflammatory processes do not resolve, perpetual platelet stimulation by pathogens or their molecular products may lead to pathological platelet hyperactivation and clotting sequelae. Pro-inflammatory processes and chronic inflammation alone can also prompt the coagulation system to take on a hypercoagulable state [39,40,41,42,43,44,45]. 

In this study, we investigate if fibrinaloid microclots are present in ME/CFS plasma and to what extent. We also analyze platelet activity, assess the amyloid load of thrombin-induced PPP clots, and measure the viscoelastic properties of blood to determine the state of coagulability (hypocoagulable vs. normocoagulable vs. hypercoagulable) [46,47]. 

## 2. Results

Table 1 shows the demographics and disease scoring of the ME/CFS cohort using the International Consensus Criteria (ICC) questionnaire for ME/CFS patients [48]. The ICC questionnaire and comorbidity results are both self-reported by participants and are therefore not used in this study to infer correlations with clinical results. Both healthy and ME/CFS population age data were normally distributed and did not yield significant differences when analyzed with an unpaired *t*-test (*p* = 0.54). Comorbidities are present within the ME/CFS population: 40% of subjects are afflicted with gastrointestinal issues; 16% are afflicted with POTS, psoriasis, fibromyalgia, gingivitis/periodontitis, hypercholesterolemia, and hypertension; 12% are afflicted with rheumatoid arthritis and cardiovascular disease; and 4% are afflicted with orthostatic hypotension, mast cell activation syndrome, rosacea, and dysautonomia. The ICC questionnaire results indicate that this study’s ME/CFS population predominantly experiences symptoms related to post-exertional neuroimmune exhaustion (7.8 ± 1.6), with subjects scoring the least in the ‘immuno, gastrointestinal, and genitourinary impairments’ section (5.9 ± 2.7). Scores that are registered as severe (i.e., with a score of 8–10) are also depicted. In total, 60% of the ME/CFS subjects experience post-exertional neuroimmune exhaustion in a severe manner; 44% and 45% experience severe neurological and energy production/transportation symptoms, respectively; and 35% report enduring severe immuno, gastrointestinal, and genitourinary impairments. Our ME/CFS population therefore constitutes a sub-population suffering the most from post-exertion-related symptoms.

Thromboelastography (TEG^®)^ analyses of whole blood (WB) and PPP is shown in Table 2. Data obtained from the analysis of ME/CFS WB were assessed against a standard clinical range of values, as provided by TEG^®^ guidelines. Conversely, ME/CFS PPP samples were compared to healthy control samples. With regards to the TEG^®^ analysis of ME/CFS WB, no subjects fell outside of the normal range for the TMRTG and TTG parameters, but our data suggest that roughly half of the participants fell outside the normal ranges for R, K, α angle, MA and MRTG (see Table 2). R and K angles reflect time-dependent properties, which indicate that our ME/CFS population clots at a rate higher than what is considered normal. ME/CFS participants exhibited larger and stronger clots (MA), which formed at a rate greater than that of controls (MRTG). 

In PPP, significant differences between the control and ME/CFS groups were only observed in the α angle (**) and MRTG (***) parameters, where the ME/CFS group exhibited higher clotting values—again pointing to a hypercoagulable state. 

Figure 1A shows fluorescence micrographs of fibrinaloid microclot presence in PPP (without added thrombin) from a cohort of 25 participants with ME/CFS and 15 healthy participants. PPP smears from the control group express little ThT signal, whereas the ME/CFS smears exhibit substantial fluorescence, therefore indicating a significant load of fibrinaloid microclots in ME/CFS blood. Figure 1B shows a micrograph plate of a fibrinaloid microclot grading system that we previously developed [47]. Figure 1C represents a strip plot of the fluorescence signal of fibrinaloid microclot micrographs from the control and ME/CFS groups as mean % area of amyloid signal, with those being vaccinated against COVID-19 highlighted in a specific color. A nonparametric Mann–Whitney test was performed, which indicated a significant difference in fibrinaloid microclot presence (*p* < 0.0001) between the control (0.10 ± 0.54) and ME/CFS (1.37 ± 3.05) % area amyloid data (Figure 1D). These qualitative and quantitative data suggest that fibrinaloid burden is notably greater in this study’s ME/CFS population when compared to the controls. It is known that SARS-CoV-2 spike proteins have the capability to influence fibrinaloid microclot formation [49]—due to spike protein S1 activity. We therefore included the vaccination status of participants to assess the potential for confounding results. Figure 1C shows that this was not the case. While slightly fewer controls than ME/CFS patients had been vaccinated, there was no discernible influence of vaccination status on the % area of fibrinaloid microclots within the two groups. All participants had received their vaccines three weeks or more prior to the collection of blood for this study. 

We also studied the amyloid presence in PPP clots that were formed by adding thrombin to ThT-incubated PPP, which forms extensive fibrin networks. Figure 2A shows representative micrographs of control and ME/CFS fibrin networks. A strip plot representing the fluorescence signal as well as the COVID vaccination status of participants is depicted in Figure 2B, which again shows no impact of vaccination status on our data. Fibrin networks created from PPP of participants with ME/CFS exhibited greater fluorescence when compared to that of healthy participants, with control and ME/CFS mean fluorescent intensity valued at 0.11 ± 0.19 and 1.69 ± 1.69 (****), respectively (Figure 2C).

Platelet morphology was also studied after adding two fluorescent platelet markers, namely PAC-1 (FITC-conjugated), which recognizes glycoprotein IIb/IIIa, and CD62P (PE-conjugated), which binds P-selectin (see Figure 3A). Platelet scoring was performed as per our previously developed scoring system (see Figure 3B,C) [47]. Mean platelet scores for the ME/CFS population are depicted in Figure 3D. The platelet populations from participants with ME/CFS exhibited a hyperactivated phenotype with significant spreading and granule release (mean spreading score of ME/CFS population: 2.72 ± 1.24). Comparably, spreading of control platelets registers with a value of 1.00 (activation with pseudopodia). In total, 80% of ME/CFS hematocrit samples exhibited hyperactivation in the context of platelet spreading, with 48% scoring 3 or 4 (severe end of platelet scoring range). Platelet clumping was observed in 52% of ME/CFS subjects, with only 32% scoring 3 or 4. Mean clumping scoring for ME/CFS population was 2.04 ± 1.21, whereas healthy controls register as ‘0’. 

## 3. Discussion

We have previously demonstrated that PPP from Long COVID/PASC individuals contains an extensive fibrinaloid microclot load and hyperactivated platelets. Due to the substantial overlap between Long COVID/PASC and ME/CFS symptomology and etiology, we performed a series of experiments to determine if fibrinaloid microclots are present in PPP from ME/CFS individuals and to what extent. We also measured viscoelastic parameters of both WB and PPP, determined the amyloid-nature of thrombin-induced fibrin networks, and assessed platelet phenotypes in hematocrit samples.

TEG^®^ analyses demonstrated that a high proportion of ME/CFS participants present with a hypercoagulable state (Table 2). In the WB analysis, several participants fell out of the healthy range, tipping towards the hypercoagulable side of the scale. This was noted in all parameters except TMRTG and TGG. In PPP, significant differences were identified in only the α angle (**) and MRTG (***), although mean ME/CFS PPP values for all TEG^®^ parameters assessed also leaned towards the hypercoagulable end. It should be noted that the extent of hypercoagulability of the ME/CFS group was not as severe as we have previously reported in diabetes [40], acute SARS-CoV-2 infection [45], or Long COVID/PASC [26], and some patients exhibited hypercoagulability whilst others did not. It is therefore worth testing for hypercoagulability in a patient-specific manner. 

Fluorescence microscopy identified fibrinaloid microclots within PPP from ME/CFS individuals with a burden significantly greater than that of controls (Figure 1). However, we note that the extent of fibrinaloid microclot load as judged by % area is significantly lower than those seen in acute COVID-19, Long COVID/PASC, and even in type 2 diabetes [25]. A comparison of comorbidities, sensitivity, and specificity is beyond the scope of the present pilot study, which was simply designed to assess whether those with ME/CFS differed in their coagulation sequelae from age- and gender-matched controls. Fibrinaloid microclots were probed for without the addition of thrombin, thereby indicating that these clot particles are circulating through the blood of afflicted individuals. Furthermore, these fibrinaloid microclots are amyloid in nature (as inferred by ThT staining) and, as such, have been shown to resist degradation via fibrinolytic means [25,42,49,50]. These fibrinaloid microclots might contribute to poor blood circulation and perfusion, possibly by blocking microcapillaries [26]. This may impact fatigue and other symptoms experienced by individuals with ME/CFS. Furthermore, fibrinaloid microclots might contribute to inflammation in the hematological system and at the endothelial linings of blood vessels in a feed-forward fashion.

A previous study demonstrated a hypercoagulable state in ME/CFS patients [30], where researchers also alluded to the role of fibrin (or fibrinaloid microclot) deposition on the endothelium and its contribution to poor perfusion and possibly the manifestation of ME/CFS symptoms. Additionally, healthy endothelial cells exposed to ME/CFS plasma exhibit functional defects [34], perhaps due to the presence of and interaction with fibrinaloid microclots. Our results are in accord with these previous findings but present an expanded and more detailed view of coagulopathies in ME/CFS. Further research regarding fibrinaloid microclots and their implication at the endothelium in the context of tissue perfusion is warranted.

Extensive fibrin clot networks (stained with ThT) were also induced where PPP samples were exposed to thrombin, and fibrin clots were assessed for the presence and load of amyloid fibrin(ogen) (Figure 2). Note that this is different from the fibrinaloid analysis as we added thrombin to form a clot network instead of probing for fibrinaloid microclots in the absence of exogenous thrombin—this gives us insight into the molecular profile of fibrin networks which mimic the terminal stages of the coagulation cascade. Amyloid protein changes were markedly increased in thrombin-induced fibrin clots from the ME/CFS group compared to controls. It is expected that clots high in amyloid content will linger in circulation and at the endothelium longer than usual, particularly due to the fact that amyloid fibrinogen is somewhat resistant to fibrinolysis [49]. Therefore, fibrinaloid microclots as well as clots formed from the terminal stages of the coagulation cascade (particularly for hemostasis) may both contribute to poor circulation and/or perfusion. 

Platelet morphology in the ME/CFS population varied across subjects, with 48% and 32% scoring as severe (3 or 4) for spreading and clumping parameters, respectively (Figure 3). It is worth acknowledging that not all participants possessed platelets indicative of a hyperactivated phenotype. It is therefore essential to apply a personalized approach to diagnostic and treatment procedures in this context of platelet activity, as well as in the assessment of hypercoagulability. 

Together, our results indicate that clotting abnormalities are present in PPP of ME/CFS individuals. It must be noted that there are past studies that have failed to show defects in elements of the coagulation system in ME/CFS [27,28], which is in some contrast to this study’s results. Importantly, however, methods such as those used in this study—including TEG and fibrinaloid microclot analysis—have not yet been applied to an ME/CFS population. It may be the case that fibrinaloid microclots and hypercoagulability have always been present in ME/CFS but just never assessed for in a reliable manner. Further testing with the TEG^®^ and probing for fibrinaloid microclots in other ME/CFS cohorts is required to corroborate the present findings and to determine if coagulopathies are widespread in the ME/CFS patient population.

As a result of this study’s findings, it is now of interest to determine if viruses implicated in ME/CFS [11], or the proteins they create, have the capacity to catalyze fibrinaloid microclot formation, as we showed that the S1 subunit of the SARS-CoV-2 spike protein is capable of doing in vitro in Long COVID/PASC PPP [49]. If ME/CFS-implicated viruses or their proteins are found capable of inducing fibrinaloid microclot formation, it will give weight to the idea that viral-induced coagulation dysfunction is a significant mechanism underlying both ME/CFS and Long COVID/PASC pathology and possibly lead to novel treatment strategies. Additionally, a novel mechanism of viral activity in ME/CFS would be established. 

More studies are required to determine what factors, including possible coagulation-promoting genetic variants or lifestyle/ environmental issues, may contribute to clotting/platelet activity in ME/CFS subsets. It is possible that dysregulated clotting, the presence of fibrinaloid microclots, and platelet abnormalities may contribute to ME/CFS symptoms. Overzealous clotting leads to hypoperfusion of the vascular system, resulting in hypoxia. Additionally, associated inflammation damages the vasculature, further exacerbating circulatory and perfusion issues. This might contribute to ME/CFS symptoms including fatigue, exercise intolerance, and cognitive impairment. 

This study utilized a variety of techniques to study different aspects of the coagulation system, thereby providing important data that can be used in both the research and clinical sector to further our understanding of ME/CFS. The relatively small sample size and self-reporting of symptom severity and comorbidities by the ME/CFS population are considered limitations of the present study. 

## 4. Materials and Methods

### 4.1. Ethical Statement

Ethical clearance for the study was obtained from the Health Research Ethics Committee (HREC) of Stellenbosch University (Stellenbosch, South Africa) ((N19/03/043, project ID #9521; yearly reapproval). For the ME/CFS and healthy volunteers who provided blood samples, the experimental objectives, risks, and details were explained, and informed consent was obtained prior to blood collection. Strict compliance to ethical guidelines and principles of the Declaration of Helsinki, South African Guidelines for Good Clinical Practice, and Medical Research Council Ethical Guidelines for Research were kept for the duration of the study and for all research protocols.

### 4.2. Sample Demographics and Blood Collection

Blood samples were obtained from healthy individuals (n = 15; 9 females, 6 males) to serve as controls for comparison. Healthy volunteers were only included if they did not smoke, did not suffer from cardiovascular disease or any coagulopathies, were not pregnant or taking contraceptives, or did not suffer from Long COVID. ME/CFS patients (n = 25; 20 females; 5 males) were recruited via the ME/CFS Foundation of South Africa and were only included in this study if they had not previously been infected with the COVID-19 virus. Participants had to have been diagnosed for longer than 6 months and were asked to complete an International Consensus Criteria (ICC) [48] based questionnaire to gain an understanding of their perspective of disease severity. We used a severity scale based on the International Consensus Criteria that converts each criterion to a 1–10 Likert severity scale item (courtesy of Michael VanElzakker Ph.D.). Blood was collected in citrated tubes by a qualified phlebotomist. Whole blood (WB) was used for viscoelastic studies, after which the samples were centrifuged at 3000× *g* for 15 min at room temperature to collect platelet-poor plasma (PPP). Platelets were identified in the hematocrit after PPP was removed and stored at −80 °C for later analysis.

### 4.3. Viscoelastic Analysis

Clotting properties of both WB and PPP samples were measured by using the Thrombelastograph^®^ (TEG^®^) 5000 Hemostasis Analyzer (Haemoscope Corp). Analyzing WB can allow for the detection of clotting abnormalities influenced by blood as a whole, while TEG^®^ of PPP allows for the assessment of the contribution of only the clotting proteins without cellular components (such as erythrocytes and platelets) [51]. Briefly, 20 μL of 0.01 M calcium chloride (required to initiate coagulation in blood drawn within citrate tubes) was added to the TEG^®^ cup, followed by 340 μL of either WB or PPP. The test was promptly started and left to run until the maximal amplitude of the clot had been reached. 

### 4.4. Fibrinaloid Microclot Analysis of PPP 

PPP was used to study fibrinaloid microclot presence in participants with ME/CFS, which was compared to healthy participants. PPP samples were incubated with the fluorescent probe, Thioflavin T (ThT) (Sigma-Aldrich, St. Louis, MO, USA), at a final concentration of 0.005 mM and for a period of 30 min, prior to viewing with a fluorescence microscope. ThT binds to open hydrophobic areas on fibrinogen, which is indicative of amyloid protein changes [50,52,53,54]. Samples were viewed on the Zeiss Axio Observer 7 fluorescent microscope with a Plan-Apochromat 63 ×/1.4 Oil DIC M27 objective (Carl Zeiss Microscopy, Munich, Germany), with the excitation wavelength set at 450–488 nm and the emission wavelength set at 499–529 nm. The % area of the fibrinaloid microclot presence (% area amyloid) was determined by analyzing micrographs using ImageJ 1.53e. After setting the scale, images are converted to 8-bit. The threshold, using the Huang setting, was then set by increasing the white background intensity to 255 and the black (fluorescence) signal intensity to 13–17. Next, we used the ‘analyze particles’ assessment and set particle size at 1-infinity. Three representative images were chosen per subject. The data generated were then analyzed with GraphPad Prism 8.4.3.

### 4.5. Amyloid Analysis of PPP Clots after Addition of Thrombin to Create Fibrin Networks

ThT was used again to identify amyloid presence and load within clot networks formed with exogenous thrombin. Briefly, 49 μL of PPP was incubated with ThT (again at a final exposure concentration of 0.005 mM) for 30 min at room temperature, and 5 μL of the sample was then transferred to a glass slide, followed by 2.5 μL of thrombin (7 U.mL^−1^, South African National Blood Service). The sample was left to stand for 2 min to allow the fibrin networks to form, after which a coverslip was placed on top of the clot. Samples were viewed on the Zeiss Axio Observer 7 fluorescent microscope with a Plan-Apochromat 63 ×/1.4 Oil DIC M27 objective (Carl Zeiss Microscopy, Munich, Germany), with ThT’s excitation wavelength set at 450–488 nm and the emission wavelength set at 499–529 nm. Fluorescent intensity of fibrin clot micrographs was calculated using ImageJ 1.53e. ‘Mean Gray Value’ and ‘area’ were chosen as measurement settings. 

### 4.6. Assessment of Platelet Phenoptype

Two fluorescent markers, PAC-1 (FITC-conjugated) (340,507, BD Biosciences, San Jose, CA, USA) and CD62P (PE-conjugated) (IM1759U, Beckman Coulter, Brea, CA, USA), were obtained to assess the state of platelets by using the Zeiss Axio Observer 7 fluorescent microscope with a Plan-Apochromat 63 × /1.4 Oil DIC M27 objective (Carl Zeiss Microscopy, Munich, Germany). After centrifugation of the blood tubes and removal of the plasma, 20 μL of hematocrit was slowly (due to its viscosity) pipetted and transferred to an Eppendorf microcentrifuge tube. After allowing the fluorescent markers to reach room temperature, 4 μL of both PAC-1 and CD62P was added to the microcentrifuge tube containing 20 μL of hematocrit. The incubation period lasted for 30 min in a dark room at room temperature. The excitation wavelength for PAC-1 was set at 450–488 nm and the emission wavelength at 499–529 nm; the excitation wavelength for CD62P was set at 540–570 nm and the emission wavelength at 577–607 nm. Platelet phenotypes were assessed with the grading system we have recently implemented [47], where platelet spreading and clumping characteristics are qualitatively analyzed in order to allocate a quantitative score of 1–4 (4 represents severe activation, whereas 1 is representative of healthy controls). A group value is determined by calculating the mean score for all images within a group, i.e., ME/CFS or control.

### 4.7. Statistical Analysis

Statistics were completed on GraphPad Prism 9.3.1. Data were subjected to normality tests (Shapiro–Wilks). Parametric data were then subject to unpaired t-tests, and non-parametric data were analyzed with the unpaired Mann–Whitney test. Data are represented as mean ± standard deviation, or median [Q1–Q3].

## 5. Conclusions

Fibrinaloid microclots and associated coagulation issues are present in ME/CFS and point to a systemic vascular pathology and potential endothelial inflammation. Although the number of participants was low, these results pave the way for larger studies. Targeted therapies to address vascular and endothelial pathology might benefit relevant individuals, although further corroboration and expansion of the present findings are required before clinical commitment. Treating apparent vascular pathology has the potential to result in the amelioration of fatigue and other ME/CFS symptoms, as it did in a Long COVID/PASC population where symptom alleviation coincided with a decrease in fibrinaloid microclot load [26,47]. It would also be of great importance to investigate the use of an algorithmic approach to hypercoagulability testing, as such an approach could potentially provide the ability to tailor diagnosis and assay selection [55]. However, it is important to note that the fibrinaloid burden in ME/CFS seems to be less than that present in Long COVID/PASC [25], and hence, caution is required when anticoagulant therapy is considered in ME/CFS. These results may pave the way for randomized control trials to investigate possible novel treatment options.

## Figures and Tables

**Figure 1 pharmaceuticals-15-00931-f001:**
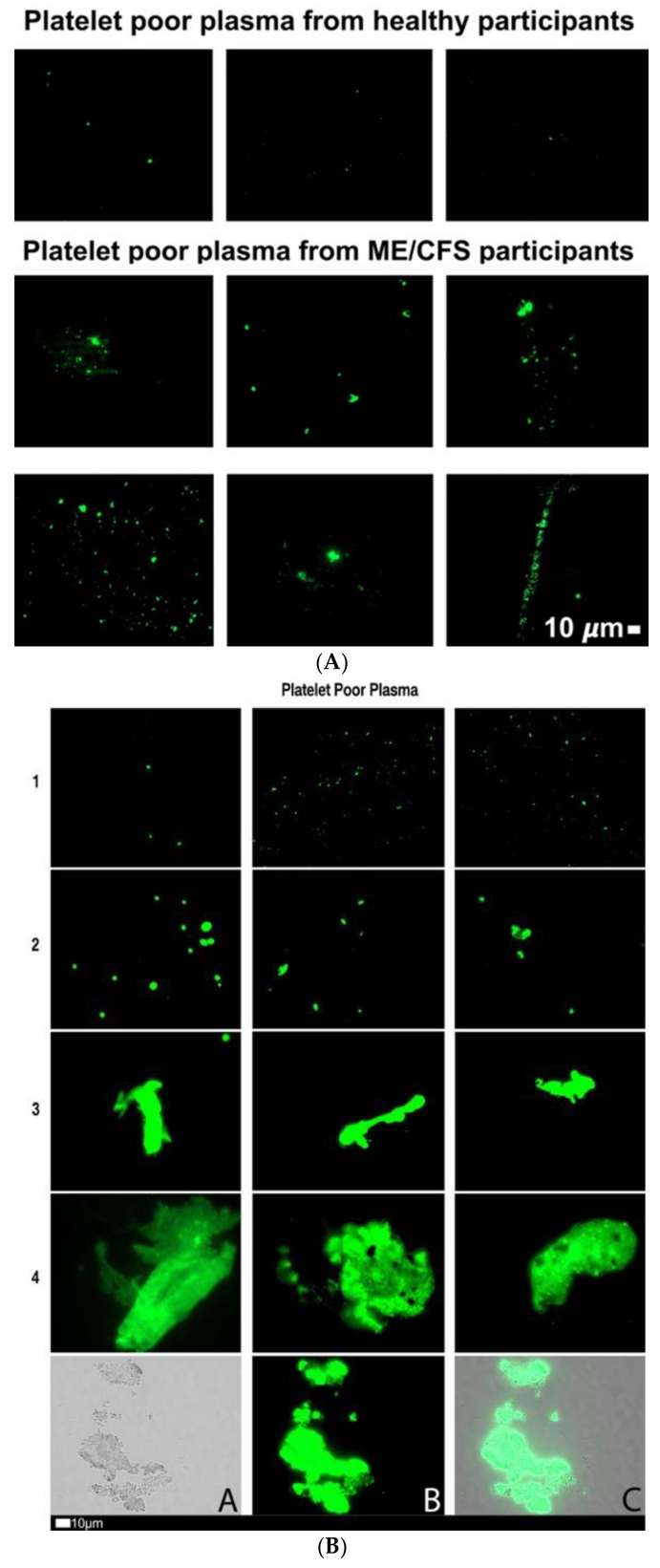
(**A**) Representative fluorescence micrographs of fibrinaloid microclot presence in PPP from controls and individuals with ME/CFS. Each image represents an individual sample. Images were taken at 63× machine magnification. (**B**) Fluorescence micrographs showing the fibrinaloid microclot scoring system used to define the load of fibrinaloid microclot severity in platelet-poor plasma (PPP). This scoring system was developed to assign a quantitative score to qualitative data. Stage 1 shows minimal fibrinaloid microclot formation as seen in healthy/control PPP, whereas Stage 4 represents a severe fibrinaloid microclot load. The bottom row represents examples of stage 4 microclots using (micrograph A) bright-field microscopy, (micrograph B) fluorescence microscopy, and (micrograph C) an overlay of fluorescence and bright-field microscopy [45]. All images within a row are a reflection of the same score. (**C**) Mean % area of amyloid signal between control and ME/CFS groups represented as a strip plot. The COVID vaccination status of individuals is color-coded and vaccinated participants are indicated with a blue block and those not vaccinated with an orange circle. (**D**) Mean % area amyloid/fibrinaloid signal between control and ME/CFS groups. A Mann–Whitney analysis yielded a significant difference (****) (*p* < 0.0001), with the ME/CFS group exhibiting a greater mean (1.37) than that of the controls (0.10). Statistical significance was recorded at *p* < 0.05. (* = *p* < 0.05; ** = *p* < 0.01; *** = *p* < 0.001).

**Figure 2 pharmaceuticals-15-00931-f002:**
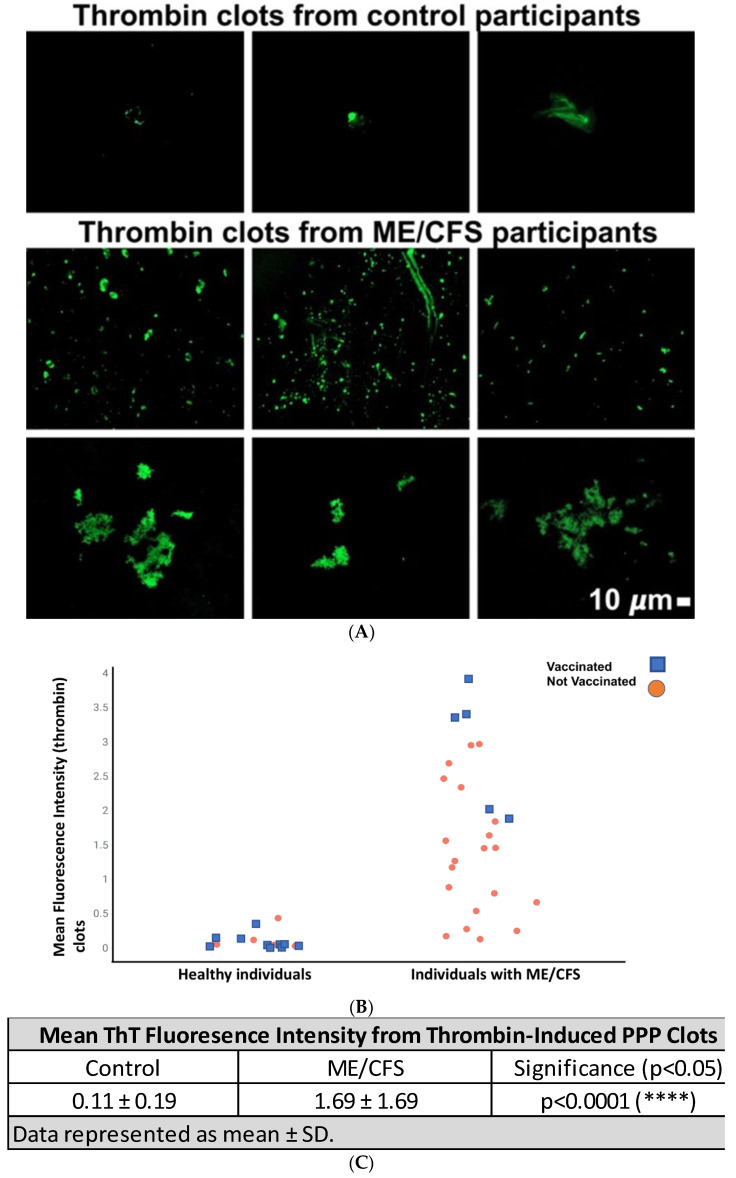
(**A**) Representative micrographs showing thrombin-induced fibrin networks stained with ThT from healthy participants and participants with ME/CFS. Each micrograph represents an individual sample. Images were taken at 63× machine magnification. (**B**) Strip plot showing the difference in mean fluorescence signal between control (0.11 ± 0.19) and ME/CFS (1.69 ± 1.69) PPP fibrin amyloid networks induced by exogenous thrombin. Vaccinated participants are indicated with a blue block and those not vaccinated, with an orange circle. (**C**) Mean fluorescent intensity of ThT signal from thrombin-induced PPP clots. A significant difference (****) between ME/CFS (1.69 ± 1.69) and controls (0.11 ± 0.19) was determined by a Mann–Whitney test. Statistical significance was recorded at *p* < 0.05.

**Figure 3 pharmaceuticals-15-00931-f003:**
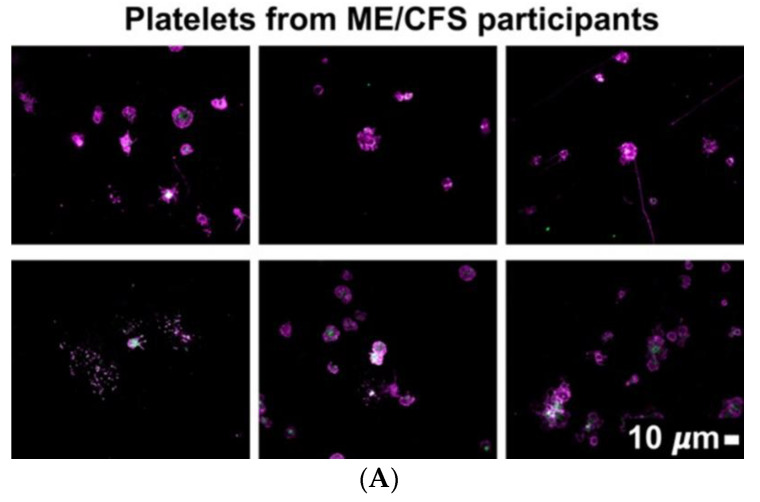
(**A**) Representative fluorescent micrographs of hematocrit samples from ME/CFS individuals stained with PAC-1 (green fluorescence) and CD62P-PE (purple fluorescence); white areas represent overlap of the two markers. Images were taken at 63× magnification. (**B**) Fluorescence micrograph examples depicting the scoring system used to measure platelet activation determined by spreading, that was previously developed to score platelet activation in Long COVID/PASC patients. This scoring system functions to assign a quantitative score to qualitative data. In Stage 1 (activation with pseudopodia), platelets are minimally activated and are seen as small and round with few pseudopodia (representative of healthy/control platelets). Severe platelet activation, characterized by large, egg-shaped morphology and aggregations, is registered as Stage 4 [47] and is indicative of hyperactivated platelets. All images within a row reflect the same score. (**C**) Fluorescence microscopy examples of the scoring system used to assess platelet clumping. This scoring system was developed to assign a quantitative score to qualitative data. Clumping is absent in Stage 1 (representative of control samples). Conversely, severe platelet clumping is registered as Stage 4 [47]. All images within a row reflect the same score. (**D**) Mean quantitative scores of platelet spreading and clumping parameters in the ME/CFS group. Platelets were scored according to the scoring system depicted in Figure 3B,C.

**Table 1 pharmaceuticals-15-00931-t001:** Demographics and scoring analysis of the ME/CFS cohort using the International Consensus Criteria (ICC) questionnaire. Both the ICC questionnaire and comorbidity results are self-reported by the ME/CFS participants. Score averages per ICC questionnaire section are given in bold face. Statistical significance was recorded at *p* < 0.05.

**Demographics**		
*p* value (parametric analysis)	0.54	
Age of control population (*n* = 15; 9 females; 6 males)	45.7 ± 6.9	
Age of ME/CFS population (*n* = 25; 20 females; 5 males)	48.2 ± 14.1	
**Comorbidities of ME/CFS Population**		
Comorbidity	% Prevalence	
Gut Dysbiosis	40%	
POTS	16%	
Fibromyalgia	16%	
Psoriasis	16%	
Gingivitis/Periodontitis	16%	
Hypercholesterolemia	16%	
Hypertension	16%	
Rheumatoid Arthritis	12%	
Cardiovascular Disease	12%	
Orthostatic Hypotension	4%	
Mast Cell Activation Syndrome	4%	
Rosacea	4%	
Dysautonomia	4%	
**ICC Questionnaire Results**		
Parameter	Average Score (out of 10)	% of Subjects Experiencing Severely (Score of 8–10)
**1. Post-Exertional Neuroimmune Exhaustion**	**7.8 ± 1.6**	**60%**
a. Marked, rapid physical and/or cognitive fatigability in response to exertion	7.9 ± 1.6	64%
b. Post-exertional symptom exacerbation (worsening of other symptoms)	7.8 ± 1.4	56%
c. Post-exertional exhaustion	8.1 ± 1.3	64%
d. Recovery period is prolonged	7.4 ± 2.0	48%
e. Low threshold of physical and mental fatigability (lack of stamina)	8.1 ± 1.7	68%
**2. Neurological Impairments**	**6.6 ± 2.6**	**44%**
a. Difficulty processing information	6.52 ± 2.00	32%
b. Short-term memory loss	6.64 ± 2.04	44%
c. Headaches	6.00 ± 3.27	40%
d. Significant pain (in muscles, tendons, abdomen, or chest)	6.68 ± 2.84	52%
e. Disturbed sleep pattern (from the previous night, e.g., insomnia, sleeping most of the day and being awake most of the night)	6.72 ± 2.82	48%
f. Unrefreshing sleep (from the previous night)	7.92 ± 2.08	60%
g. Neurosensory and perceptual symptoms (e.g., inability to focus vision, sensitivity to light, noise, etc.)	7.20± 1.87	56%
h. Motor symptoms (e.g., twitching, poor coordination)	5.12 ± 2.65	16%
**3. Immuno, Gastrointestinal, and Genitourinary Impairments**	**5.9 ± 2.7**	**35%**
a. Flu-like symptoms (e.g., sore throat, sinusitis, enlarged or tender lymph nodes)	5.48 ± 1.94	16%
b. Gastrointestinal tract symptoms (e.g., nausea, abdominal pain, bloating, irritable bowel)	6.32 ± 2.66	44%
c. Genitourinary symptoms (e.g., urinary urgency, urinary frequency, nocturia or having to urinate two or more times a night)	5.76 ± 2.96	32%
d. Sensitivities to food, medications, odors, or chemicals	6.16 ± 3.05	48%
**Energy Production/Transportation Impairments**	**6.6 ± 2.6**	**45%**
a. Cardiovascular symptoms (e.g., orthostatic intolerance, postural orthostatic tachycardia syndrome or POTS, palpitations, light-headedness/dizziness)	7.00 ± 2.38	52%
b. Respiratory symptoms (e.g., air hunger, labored breathing, fatigue of chest wall muscles)	5.64 ± 2.98	32%
c. Loss of thermostatic stability (e.g., subnormal body temperature, sweating episodes, recurrent feelings of feverishness, cold extremities)	7.28 ± 1.54	52%
d. Intolerance of extremes of temperature	6.44 ± 2.89	44%
Data are represented as mean ± standard deviation.		

**Table 2 pharmaceuticals-15-00931-t002:** TEG^®^ analysis recorded in both whole blood (WB) and platelet-poor plasma (PPP). ME/CFS WB samples were assessed against a clinical standard, whereas ME/CFS PPP samples were assessed alongside control PPP.

**TEG® of WB**			
Parameter	Standard WB Range	ME/CFS WB (*n* = 25)	Out of Standard Range
R	9–27	7.73 ± 2.31	16/64% (below)
K	2–9	2.39 ± 0.82	9/36% (below)
α-Angle	22–58	56.99 ± 10.25	13/52% (above)
Maximum Amplitude	44–64	65.64 ± 7.02	14/56% (above)
MRTG	0–10	10.05 [7.18–12.12]	13/52% (above)
TMRTG	5–23	9.74 ± 2.73	0
TTG	251–1041	787.50 ± 84.07	0
**TEG® of PPP**			
Parameter	Control PPP (*n* = 15)	ME/CFS PPP (*n* = 25)	*p* value
R	10.25 ± 1.97	8.54 ± 3.51	0.09
K	3.10 [2.80–3.80]	1.90 [1.40–4.75]	0.05
α-Angle	51.70 [45.00–57.10]	61.80 [54.00–66.25]	0.002 (**)
Maximum Amplitude	25.90 [23.00–27.10]	29.30 [22.05–32.55]	0.20
MRTG	8.05 [6.80–8.93]	12.17 [9.13–15.77]	0.0009 (***)
TMRTG	11.30 ± 2.20	9.68 ± 3.71	0.13
TTG	306.40 [283.30–324.10]	350.90 [261.30–391.00]	0.22

Data are represented as mean ± standard deviation, or median [Q1–Q3]. The standard WB range for the various parameters is as indicated by TEG® guidelines, and is used as a clinical standard to assess the state of coagulation in ME/CFS WB samples. Statistical significance was recorded at *p* < 0.05 (** = *p* < 0.01; *** = *p* < 0.001).

## Data Availability

Data is contained within the article.

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
