# Peer review of "The Occurrence of Hyperactivated Platelets and Fibrinaloid Microclots in Myalgic Encephalomyelitis/Chronic Fatigue Syndrome (ME/CFS)"

_pharmaceuticals, 2022, doi:10.3390/ph15080931_

Round 1

Reviewer 1 Report

This paper applies findings from and techniques used in a new medical condition, long COVID, to a mysterious illness, myalgic encephalomyelitis/ chronic fatigue syndrome (ME/CFS),  that often starts after an infection, shares a similar set of disabling symptoms without a clearly known mechanism, but has been present for decades. I anticipate that this publication will be received with much interest.

The authors tested for hypercoagulability in samples of whole blood and platelet poor plasma using 4 different techniques: assays to test for viscoelasticity, microscopic examination of platelet poor plasma (PPP) as-is for microclots, examination of PPP with addition of thrombin, and evaluation of platelet clumping with fluorescent platelet markers. Overall, they found the blood of ME/CFS patients favored coagulation compared to that of healthy controls but contained fewer microclots than that of long COVID subjects. 

My main suggestions concern a) making the graphics more understandable and b) detailing in the Discussion how this work fits in with prior studies, the strengths/ limitations of the current research, and what needs to be done next to advance progress. 

METHODS:

1. I understand this is not a case-control study but the healthy controls and ME/CFS participants were selected using slightly different criteria other than presence of ME/CFS. For example, controls were required to have no history of coagulopathies whereas the sick participants could not have evidence of SARS-CoV-2 infection. 

2. In Table 1, the co-morbidities of ME/CFS  participants are listed. Were these based on patient self-report? Self-report can be inaccurate with participants both over- and under-reporting diagnoses as well as giving diagnoses which are inaccurate. Additionally, "leaky gut" is a very vague term. Some participants may have gut dysbiosis confirmed by a healthcare professional with tests showing abnormalities but other participants might have claimed this diagnosis for themselves based on any presence of gut symptoms. 

How might these differences between the 2 groups and self-report of co-morbidities affect the Results and Conclusions? What can be done in the future to address any possible effects?

GRAPHICS:

1. Table 2: Was TEG analysis also performed on the whole blood of controls or was the comparison performed only against laboratory reference standards? If so, it would be helpful (as in the lower part of Table 2) to show the results of the controls. Rather than giving only the number of sick participants with values outside the normal range, give both percentage and number.

2. Figures 1B, 4B, and 4C: It is not clear to me what these figures and numbers assigned to them were meant to show. Was the intention to demonstrate how a quantitative score was assigned to a qualitative image? Additionally, these figures consisted of 3 X 4 images. Within rows, e.g. 3 images in row with "3" assigned, were all 3 images give a score (Stage) of 3 or are they assigned different scores? When a number like "2.72" or "2.04" is noted, is that the mean score of the images within a group? A bit more explanation to the reader would help.

3. Fig 1A and Fig 3: Does each image in these figures representing sick or healthy groups come from a single individual or did some images come from the same individual? How samples are chosen can affect the result. 

4. The way results was sprinkled in the text between figures was confusing and difficult to read. It might be helpful were there another table summarizing the data, e.g. mean fluorescent intensity, % with Stage 3 or 4 platelet clumping, etc. between the 2 groups. 

5. Fig 2: The authors can consider whether using different symbols (e.g. dots for unvaccinated, crosses for vaccinated) to denote vaccination status might be easier to understand for some readers rather than different colors. Originally, I printed this out to read on a black-and-white printer and it was hard to see the difference. The same issue might affect people who are color-blind. 

DISCUSSION: 

1. Context of work: In the Introduction, the authors cited a number of publications (references 27-34) that suggest platelet/ endothelial dysfunction and/or hypercoagulability in ME/CFS participants in the past. The Discussion should circle back to these or other studies and elaborate on how this work fits in with the prior work.

For example: How do these results compare with the prior? Do they support or contradict prior results? Or maybe support some parts but not others? There are some tests clinicians can readily order to test for hypercoagulability (examples in citation below): were these done in the prior studies? If they were done and showed normal results, how do the assays the authors use differ from these tests? If they were not performed, should they be?

2. Strengths/ Weaknesses/ Limitations: What are the major strengths of this study, e.g. using a variety of tests to examine the issue of hypercoagulability, microclots? A major limitation would be the small sample size. 

3. Future studies: Given the findings of this paper, what do the authors plan to do next? What are areas that researchers should explore further? What can be done to strengthen the findings or address/ allay any weaknesses/ limitations identified? Some examples might be replicating the study in a large sample, testing subjects for or asking them about other hypercoagulable disorders/ symptoms, etc. 

Citation:

Nakashima MO, Rogers HJ. Hypercoagulable states: an algorithmic approach to laboratory testing and update on monitoring of direct oral anticoagulants. Blood Res. 2014 Jun;49(2):85-94. doi: 10.5045/br.2014.49.2.85. Epub 2014 Jun 25. PMID: 25025009; PMCID: PMC4090343.

Author Response

Dear reviewer 1, please find our amendments as per your suggestions.  We have added comments below each of your suggestions:

This paper applies findings from and techniques used in a new medical condition, long COVID, to a mysterious illness, myalgic encephalomyelitis/ chronic fatigue syndrome (ME/CFS),  that often starts after an infection, shares a similar set of disabling symptoms without a clearly known mechanism, but has been present for decades. I anticipate that this publication will be received with much interest.

Thank you for your comment!

The authors tested for hypercoagulability in samples of whole blood and platelet poor plasma using 4 different techniques: assays to test for viscoelasticity, microscopic examination of platelet poor plasma (PPP) as-is for microclots, examination of PPP with addition of thrombin, and evaluation of platelet clumping with fluorescent platelet markers. Overall, they found the blood of ME/CFS patients favored coagulation compared to that of healthy controls but contained fewer microclots than that of long COVID subjects. 

My main suggestions concern a) making the graphics more understandable and b) detailing in the Discussion how this work fits in with prior studies, the strengths/ limitations of the current research, and what needs to be done next to advance progress. 

METHODS:

  1. I understand this is not a case-control study but the healthy controls and ME/CFS participants were selected using slightly different criteria other than presence of ME/CFS. For example, controls were required to have no history of coagulopathies whereas the sick participants could not have evidence of SARS-CoV-2 infection. 
  2. In Table 1, the co-morbidities of ME/CFS participants are listed. Were these based on patient self-report? Self-report can be inaccurate with participants both over- and under-reporting diagnoses as well as giving diagnoses which are inaccurate. Additionally, "leaky gut" is a very vague term. Some participants may have gut dysbiosis confirmed by a healthcare professional with tests showing abnormalities but other participants might have claimed this diagnosis for themselves based on any presence of gut symptoms. 

The ICC Questionnaire and comorbidity results were self-reported. We have now mentioned so in the paper: ‘The ICC questionnaire and comorbidity results are both self-reported by participants, and are therefore not used in this study to infer correlations with clinical results’. Leaky gut’ has also been removed from Table 1.

How might these differences between the 2 groups and self-report of co-morbidities affect the Results and Conclusions? What can be done in the future to address any possible effects?

We have stated that we are not utilizing the self-reported data to infer correlations between clinical results. It is simply there to aid in optimizing the description of our population.

GRAPHICS:

  1. Table 2: Was TEG analysis also performed on the whole blood of controls or was the comparison performed only against laboratory reference standards? If so, it would be helpful (as in the lower part of Table 2) to show the results of the controls. Rather than giving only the number of sick participants with values outside the normal range, give both percentage and number.

We have stated in the caption of Table 2: ‘ME/CFS WB samples were assessed against a clinical standard, whereas ME/CFS PPP samples were assessed alongside control PPP.’ We have also added the percentage next to the no. of participants outside the normal range.

  1. Figures 1B, 4B, and 4C: It is not clear to me what these figures and numbers assigned to them were meant to show. Was the intention to demonstrate how a quantitative score was assigned to a qualitative image? Additionally, these figures consisted of 3 X 4 images. Within rows, e.g. 3 images in row with "3" assigned, were all 3 images give a score (Stage) of 3 or are they assigned different scores? When a number like "2.72" or "2.04" is noted, is that the mean score of the images within a group? A bit more explanation to the reader would help.

Figures have been adjusted for clarity. Figures are grouped according to experiments, and quantitative data is included. We have stated, with regards to scoring figures, what their purpose is (meant for quantitative analysis of qualitative data). For the scoring figures, we have also mentioned: All images within a row are a reflection of the same score’. For the quantitative data, we have mentioned in the caption of Figure 1D and Figure 3D: ‘Mean % area amyloid/fibrinaloid signal between control and ME/CFS groups. Mean quantitative scores of platelet spreading and clumping parameters in the ME/CFS group’.

  1. Fig 1A and Fig 3: Does each image in these figures representing sick or healthy groups come from a single individual or did some images come from the same individual? How samples are chosen can affect the result. 

We have now mentioned that each image is an individual sample. The qualitative figures contain within their caption the term ‘Representative’, which implies these are images representative of the group. The quantitative analysis was done on the group.

  1. The way results was sprinkled in the text between figures was confusing and difficult to read. It might be helpful were there another table summarizing the data, e.g. mean fluorescent intensity, % with Stage 3 or 4 platelet clumping, etc. between the 2 groups. 

Quantitative data has been added to the relevant figures.

  1. Fig 2: The authors can consider whether using different symbols (e.g. dots for unvaccinated, crosses for vaccinated) to denote vaccination status might be easier to understand for some readers rather than different colors. Originally, I printed this out to read on a black-and-white printer and it was hard to see the difference. The same issue might affect people who are color-blind. 

Unique symbols for vaccinated and unvaccinated individuals have now been used, instead of colour-coded points. We have used colour and blocks for ease.

DISCUSSION: 

  1. Context of work: In the Introduction, the authors cited a number of publications (references 27-34) that suggest platelet/ endothelial dysfunction and/or hypercoagulability in ME/CFS participants in the past. The Discussion should circle back to these or other studies and elaborate on how this work fits in with the prior work.

For example: How do these results compare with the prior? Do they support or contradict prior results? Or maybe support some parts but not others? There are some tests clinicians can readily order to test for hypercoagulability (examples in citation below): were these done in the prior studies? If they were done and showed normal results, how do the assays the authors use differ from these tests? If they were not performed, should they be?

We have now added sections of text where we circle back to the few, past studies pertinent to coagulation. We have also highlighted the distinctiveness of our techniques, and alluded to the notion that our tests are more specific/able to provide a more effective assessment than those previously employed. We also stated that more studies using our techniques - TEG® and fibrinaloid microclot detection – is required for corroboration.

  1. Strengths/ Weaknesses/ Limitations: What are the major strengths of this study, e.g. using a variety of tests to examine the issue of hypercoagulability, microclots? A major limitation would be the small sample size.

We have stated that the sample size and self-reporting assessments are limitations of our study. With regards to strengths, we have added: ‘This study utilized a variety of techniques to study different aspects of the coagulation system, thereby providing important data that can be used in both the research and clinical sector to further our understanding of ME/CFS.

  1. Future studies: Given the findings of this paper, what do the authors plan to do next? What are areas that researchers should explore further? What can be done to strengthen the findings or address/ allay any weaknesses/ limitations identified? Some examples might be replicating the study in a large sample, testing subjects for or asking them about other hypercoagulable disorders/ symptoms, etc. 

We have now stated:

As a result of this study’s findings, it is now of interest to determine whether or not the viruses previously implicated in ME/CFS 11 have the capability to induce fibrinaloid microclot formation, as does the SARS-CoV-2 virus in Long Covid/PASC.’;

More studies are required to determine what factors, including possible coagulation-promoting genetic variants or lifestyle/ environmental issues, may contribute to clotting/platelet activity in ME/CFS subsets.’;

Further testing with the TEG® and probing for fibrinaloid microclots in ME/CFS populations is required to corroborate the present findings and determine if coagulopathies are widespread within ME/CFS.’

We have added the reference below to our conclusion:

It would also be of great importance to investigate the use of an algorithmic approach to hypercoagulability testing, as such an approacy could potentially provide the ability to tailor diagnosis and assay selection55

Citation added as no 55:

Nakashima MO, Rogers HJ. Hypercoagulable states: an algorithmic approach to laboratory testing and update on monitoring of direct oral anticoagulants. Blood Res. 2014 Jun;49(2):85-94. doi: 10.5045/br.2014.49.2.85. Epub 2014 Jun 25. PMID: 25025009; PMCID: PMC4090343.

Reviewer 2 Report

Excellent working hypothesis

I think that the n should be increased and that other groups replicated the results.

In this patients with CFS and coagulation disorders, you have observed some thrombotic phenomen ?

And another serious question,  would have to be anticoagulation and antiaggregation

Author Response

Please find modifications and comments from reviewer 2:

Excellent working hypothesis

Thanks!

I think that the n should be increased and that other groups replicated the results.

We do definitely acknowledge that the n is small and added a comment to the conclusion.

Although the number of participants were low, these results pave the way for larger studies.

In this patients with CFS and coagulation disorders, you have observed some thrombotic phenomen ?

We used the self-reported symptom questionnaire, and this with their comorbidities could possibly point to thrombotic phenomenon other than what we have shown with our lab tests.

And another serious question,  would have to be anticoagulation and anti-aggregation.  

We are very cautious to speculate on this, as we are aware that patients may want to search out treatment.  Before properly controlled RCTs this is perhaps a premature speculation.  However, these results many now pave the way for such trials.